# Integrated analysis of human DNA methylation, gene expression, and genomic variation in iMETHYL database using kernel tensor decomposition-based unsupervised feature extraction

Y-h. Taguchi[1]*, Shohei Komaki[2], Yoichi Sutoh[2], Hideki Ohmomo[2], Yayoi Otsuka-Yamasaki[2], Atsushi Shimizu[2]

**1** Department of Physics, Chuo University, Tokyo, Japan, **2** Division of Biomedical Information Analysis, Iwate Tohoku Medical Megabank Organization, Disaster Reconstruction Center, Iwate Medical University, Iwate, Japan

* tag@granular.com

**Data Availability Statement:** Summary statistics of DNA methylation, gene expression, and genomic

## Abstract

Integrating gene expression, DNA methylation, and genomic variants simultaneously without location coincidence (i.e., irrespective of distance from each other) or pairwise coincidence (i.e., direct identification of triplets of gene expression, DNA methylation, and genomic variants, and not integration of pairwise coincidences) is difficult. In this study, we integrated gene expression, DNA methylation, and genome variants from the iMETHYL database using the recently proposed kernel tensor decomposition-based unsupervised feature extraction method with limited computational resources (i.e., short CPU time and small memory requirements). Our methods do not require prior knowledge of the subjects because they are fully unsupervised in that unsupervised tensor decomposition is used. The selected genes and genomic variants were significantly targeted by transcription factors that were biologically enriched in KEGG pathway terms as well as in the intra-related regulatory network. The proposed method is promising for integrated analyses of gene expression, methylation, and genomic variants with limited computational resources.

## Introduction

The integrated analysis of multiomics datasets has always been difficult; in particular, integrating gene expression, DNA methylation, and genetic variants has rarely been successful [1, 2]; in contrast, many studies integrate two of these three, that is, DNA methylation and genomic variants [3–5], gene expression and DNA methylation [6–9], and gene expression and genomic variants [10]. Although Seal et al. [2] successfully predicted gene expression from copy number variants (CNV) and DNA methylation, they did not discuss the relationship between CNV and DNA methylation. Therefore, they did not conduct a truly integrated analysis. Bell et al. [1] examined DNA methylation as a function of genetic and gene expression variation

variation are available from the NBDC database (https://humandbs.biosciencedbc.jp/en/hum0056-v1). Individual-level data cannot be made publicly available to protect the participants' privacy but is available upon approval of an application to the Tohoku Medical Megabank Project (https://www.megabank.tohoku.ac.jp/english/; http://iwate-megabank.org/en/).

**Funding:** This study was partially funded by the Tohoku Medical Megabank project, supported by the Ministry of Education, Culture, Sports, Sciences, and Technology of the Japanese government and the Japan Agency for Medical Research and Development. This study was also supported by KAKENHI, [grant numbers 19H05270, 20H04848, and 20K12067] to YHT. The super computer resource (powered by AMED research grant JP20km0405001) was provided by Tohoku Medical Megabank Organization, Tohoku University. The funders had no role in study design, data collection and analysis, decision to publish, or preparation of the manuscript.

**Competing interests:** The authors have declared that no competing interests exist.

but did not directly investigate the relationship between gene expression and genetic variants; therefore, it was not a true integrated analysis.

In this study, we applied a recently proposed method [11] for the integrated analysis of gene expression, genetic variants, and DNA methylation using data retrieved from the iMETHYL database [12, 13], without assuming any causal relationship between them in the framework of a purely data-driven strategy. Gene expression, methylation, and genetic variation shared patient-dependent patterns and were regulated by transcription factors. Enrichment analysis based on the genes targeted by these transcription factors is largely related to various biological functions.

## Materials and methods

### Data set

The data set comprised gene expression, DNA methylation, and genomic variation profiles obtained from the same patients for each cell type (CD4 positive T cells: 99 patients, monocytes: 99 patients, neutrophils: 94 patients, for a total of 194 unique subjects. Venn diagram in Fig 1); 194 subjects common among these three measurements (i.e., gene expression, DNA methylation, and genomic variation) in one of three cell types were included in the analysis. The dataset analyzed in this study was obtained from the iMETHYL database after receiving approval from the Medical Ethics Committee of Iwate Medical University (approval no. HGH29-32) and the Ethics Committee of Chuo University (2019-6 and 2021-072).

### Preprocessing

Fastq files obtained from RNA-seq were processed following the GTEx pipeline V8 [14] with slight modifications. Briefly, sequence reads were aligned to the GRCh37 human reference genome using STAR v2.5.0 [15], and bam files were generated.

Sequence reads obtained from whole-genome bisulfite sequencing were aligned using NovoAlign v3.02.08 (Novocraft Technologies, Sdn. Bhd., Selangor, Malaysia). The number of

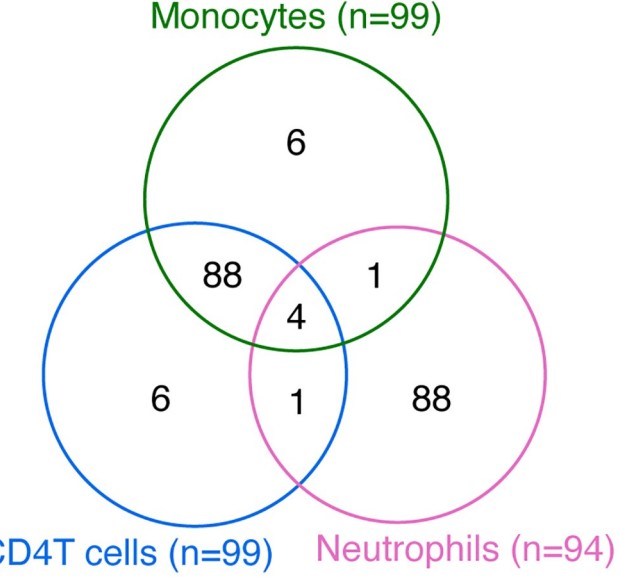

**Fig 1. Venn diagram of subjects in CD4T cells, monocytes, and neutrophils.**

converted and unconverted cytosines mapped to each CpG was counted using NovoMethyl v3.02.08 (Novocraft Technologies), and the proportion of unconverted cytosines was calculated as the DNA methylation level (%) [16].

Whole-genome sequence data were obtained from the Tohoku Medical Megabank Project [17], in which sequence reads were mapped onto the CRCh37 human reference genome using BWA-MEM [18] and variant calls were carried out using GATK v3.7 [19]. The resultant VCF files were further converted into the 012 format, where numeric variables ranging from 0 to 2 represent the number of non-reference alleles.

For the gene expression profiles, the bam files were converted into bed files using the bamtobed command. For gene expression and DNA methylation profiles, the bed files were separately integrated (summed or averaged) over every 25,000 nucleotide intervals, separately for 22 individual autosomes. Hereafter, these intervals are denoted as "genomic regions." Genetic variants were converted to numeric values (0–2) representing the number of non-reference alleles.

## Tensor decomposition-based unsupervised feature extraction

We applied TD-based unsupervised FE optimized for multiomics data integration [11] to the dataset. Suppose that $x_{i_k jk} \in \mathbb{R}^{N_k \times M \times 3}$ represents the values of the $i_k$th components of the $k$th omics dataset for the $j$th subject. From these, we generated

$$x_{jj'k} = \sum_{i_k=1}^{N_k} x_{i_k jk} x_{i_k j'k} \in \mathbb{R}^{M \times M \times 3} \tag{1}$$

HOSVD [20] was applied to $x_{jj'K}$ resulting in

$$x_{jj'k} = \sum_{\ell_1=1}^{M} \sum_{\ell_2=1}^{M} \sum_{\ell_3=1}^{3} G(\ell_1 \ell_2 \ell_3) u_{\ell_1 j} u_{\ell_2 j'} u_{\ell_3 k} \tag{2}$$

where $G(\ell_1 \ell_2 \ell_3) \in \mathbb{R}^{M \times M \times 3}$ is a core tensor that represents a weight of the product $u_{\ell_1 j} u_{\ell_2 j'} u_{\ell_3 k}$ towards $x_{jj'k}$. $u_{\ell_1 j}, u_{\ell_2 j'} \in \mathbb{R}^{M \times M}$ and $u_{\ell_3 k} \in \mathbb{R}^{3 \times 3}$ are singular value orthogonal matrices. $u_{\ell_1 j} = u_{\ell_2 j'}$ when $\ell_1 = \ell_2$ and $j = j'$. After identifying $u_{\ell_1 j}$ of interest and denoting a set of these $\ell_1$s as $\Omega_{\ell_1}$, we can derive the singular value vectors attributed to $i_k$s by

$$u_{\ell_1 i_k} = \sum_{j=1}^{M} x_{i_k jk} u_{\ell_1 j} \tag{3}$$

and attribute $P$values to $i_k$ assuming that $u_{\ell_1 i_k}$ follows a Gaussian distribution (null hypothesis)

$$P_{i_k} = P_{\chi^2} \left[ > \sum_{\ell_1 \in \Omega_{\ell_1}} \left( \frac{u_{\ell_1 i_k}}{\sigma_{\ell_1}} \right)^2 \right] \tag{4}$$

where $P_{\chi^2}[> x]$ is the cumulative $\chi^2$ distribution whose argument is larger than $x$ and $\sigma_{\ell_1}$ is the standard deviation. $P_{i_k}$s were corrected using the BH criterion [20] and $i_k$s with an adjusted $P_{i_k}$ of less than 0.01 were selected.

## Identification of genes associated with selected genomic regions

Genes included in the genomic regions selected by KTD-based unsupervised FE were identified using biomaRt [21] package in R [22] for the hg19 genome.

### Enrichment analysis

Enrichment analysis was performed using Enrichr software [23].

### Transcription factor regulation analysis

Information on TF mutual regulation relations was retrieved from Regnetworkweb [24] and TRRUST2 [25].

### Identification of TFBSs and genes associated with detected genetic variants

TFBSs and genes associated with genetic variation were identified using SNPnexus [26].

## Results

Fig 2 presents a flowchart of the analyses.

### Identification of $u_{\ell_1 j}$s of interest

Since this study included only healthy individuals, we could not identify differentially expressed genes (DEG). Therefore, we employed a fully unsupervised strategy. Defining DEG is difficult using this strategy. Our criterion was as follows: seek $u_{\ell_1 j}$s that are common among distinct individual autosomes. If the 22 $u_{\ell_s j}$s identified for individual autosomes share the same subject dependence $j$, it is unlikely to be accidental. Although there is some possibility that they reflect measurement bias; for example, if the total number of reads differs from subject to subject, this is very unlikely to be caused by measurement bias for the following reasons. First, as the present study was an integrated analysis of three omics measurements, the same patterns of subject measurement bias were unlikely to occur for the three omics datasets simultaneously because the experimental procedures differed from each other. Second, as $u_{\ell_1 j}$ are orthogonal to each other for distinct $\ell_1$s, if more than two patterns of subject dependence are observed for more than one $\ell$s, none of them can be interpreted as measurement bias, which can result in only one unique pattern of subject dependence. Third, if subject-dependent patterns are caused by measurement bias, the selected genes based on these patterns may not be biologically reasonable; however, by applying enrichment analysis, the biological significance of the selected genes is easily validated.

Table 1 lists the average absolute mutual correlation coefficients between the patterns attributed to 22 autosomes:

$$\langle \rho_{\mathrm{chr}} \rangle = \frac{1}{21} \sum_{\mathrm{chr}' \neq \mathrm{chr}} \left| \rho\left( u_{\ell_1 j}^{\mathrm{chr}}, u_{\ell_1 j}^{\mathrm{chr}'} \right) \right| \tag{5}$$

where $\rho\left( u_{\ell_1 j}^{\mathrm{chr}}, u_{\ell_1 j}^{\mathrm{chr}'} \right)$ is the correlation coefficient between $u_{\ell_1 j}^{\mathrm{chr}}$ and $u_{\ell_1 j}^{\mathrm{chr}'}$ and $u_{\ell_1 j}^{\mathrm{chr}}$ is $u_{\ell_1 j}$ selected for chr-th autosome ($1 \leq \mathrm{chr} \leq 22$). These factors are mutually correlated. To further validate the significance of the number of pairs among the total $22 \times 21/2 = 231$ pairs, we computed $P$-values and counted the number of pairs associated with significant correlations. Most pairs were significantly correlated (Table 2). Hereafter this set of $u_{\ell_1 j}$ is denoted as the "max correlated set." Next, we attempted to determine whether $u_{\ell_1 j}$s in the "max correlated set" correlated with the clinical data (Table 3). Unfortunately $\ell_1$s selected most frequently in Table 1 for "max correlated set," $u_{\ell_1 j}$s with $\ell_1 = 2$, are not correlated with clinical data. Thus, we decided to select $u_{\ell_1 j}$s with $\ell_1 = 3$ for CD4 + T cells and neutrophils as additional singular value vectors of interest (Those for monocytes were not selected because they did not correlate with the clinical

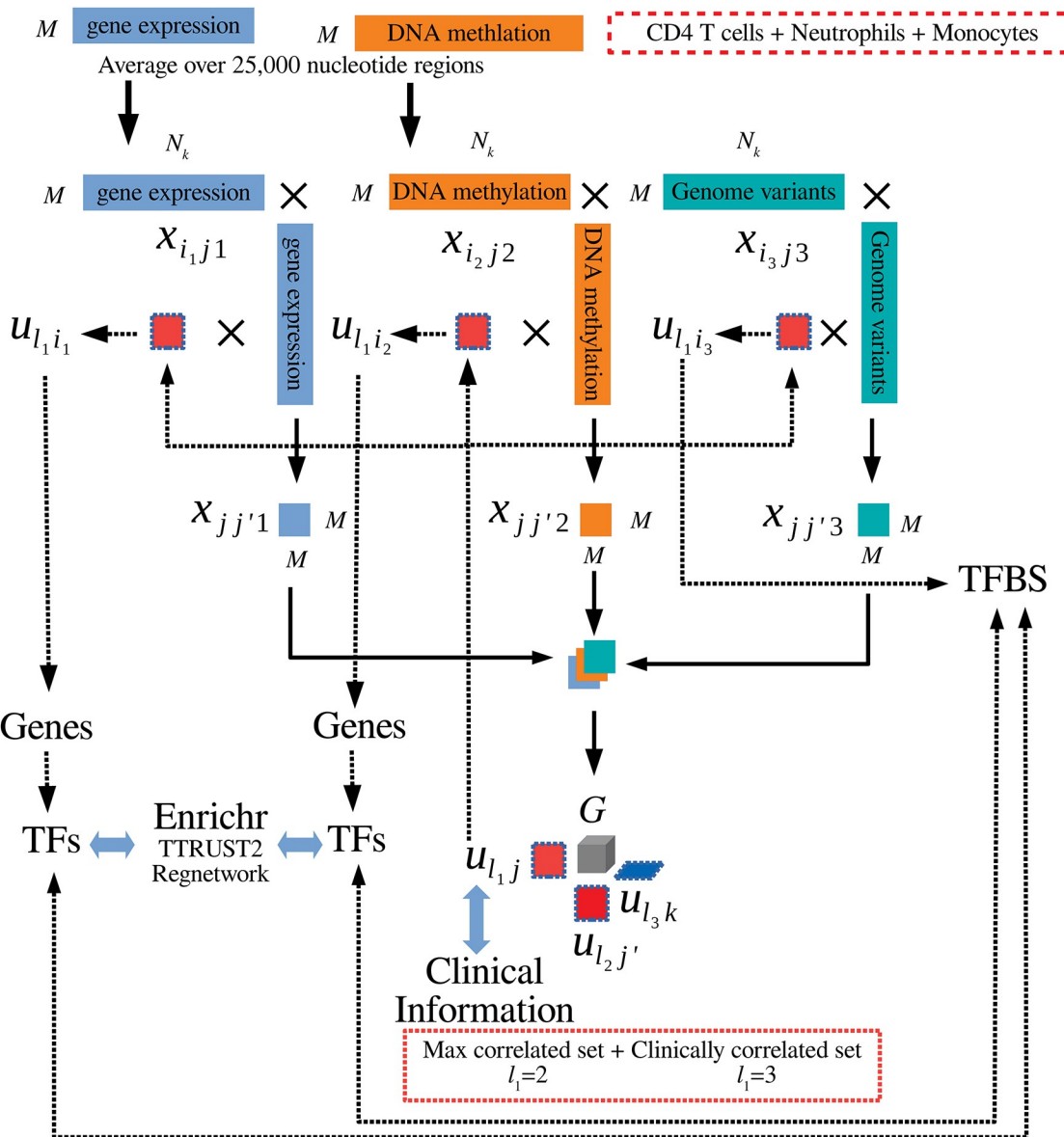

**Fig 2. Flowchart of the analyses for CD4T cells, neutrophils, and monocytes.** Gene expression and DNA methylation profiles were averaged over 25,000 nucleotide regions. Gene expression, DNA methylation and genomic variants were multiplied by themselves to obtain square matrices that are bundled into a tensor; tensor decomposition was then applied. The obtained $u_{\ell_1 j}$ were compared with clinical information and used to compute $u_{\ell_1 i_k}$ to select regions ("max correlated set" ($\ell_1 = 2$) and "clinically correlated set" ($\ell_1 = 3$) are identified). For gene expression and DNA methylation, TFs that target genes included in the identified regions were selected and validated with enrichment analyses and comparisons with TTRUST2 and Regnetwork. Identified TFs were also compared with TFBSs identified by the selected genomic variants.

data in Table 3). Table 1 also shows the mutual correlation coefficients between the patterns attributed to the 22 autosomes and the associated and corrected $P$-values for $u_{\ell_1 j}$s with $\ell_1 = 3$. Although the correlations were less than those in the "max correlated set," they were more or less significant (Table 1), as the majority of the 231 pairs were significantly correlated (Table 3). Thus, we decided to employ $u_{\ell_1 j}$s with $\ell_1 = 3$ for the downstream analyses. Hereafter, these sets with $u_{\ell_1 j}$ are denoted as "clinically correlated sets."

**Table 1. Averaged mutual correlations between $u_{\ell_1 j}$s.**

| | Max correlated set | | | | | | Clinically correlated set | | | |
|---|---|---|---|---|---|---|---|---|---|---|
| | CD4 T Cells | | Monocytes | | Neutrophils | | CD4 T Cells | | Neutrophils | |
| chr | $\ell_1$ | $\langle \rho_{chr} \rangle$ | $\ell_1$ | $\langle \rho_{chr} \rangle$ | $\ell_1$ | $\langle \rho_{chr} \rangle$ | $\ell_1$ | $\langle \rho_{chr} \rangle$ | $\ell_1$ | $\langle \rho_{chr} \rangle$ |
| 1 | 2 | 0.73 | 2 | 0.70 | 2 | 0.89 | 3 | 0.29 | 3 | 0.72 |
| 2 | 2 | 0.73 | 2 | 0.53 | 2 | 0.91 | 3 | 0.27 | 3 | 0.75 |
| 3 | 2 | 0.85 | 2 | 0.75 | 2 | 0.91 | 3 | 0.72 | 3 | 0.75 |
| 4 | 2 | 0.79 | 2 | 0.48 | 2 | 0.74 | 3 | 0.71 | 3 | 0.51 |
| 5 | 2 | 0.62 | 2 | 0.61 | 2 | 0.80 | 3 | 0.29 | 3 | 0.58 |
| 6 | 2 | 0.85 | 2 | 0.69 | 2 | 0.87 | 3 | 0.73 | 3 | 0.71 |
| 7 | 2 | 0.84 | 2 | 0.62 | 2 | 0.90 | 3 | 0.73 | 3 | 0.71 |
| 8 | 2 | 0.85 | 2 | 0.72 | 2 | 0.90 | 3 | 0.72 | 3 | 0.71 |
| 9 | 2 | 0.85 | 2 | 0.66 | 2 | 0.91 | 3 | 0.73 | 3 | 0.74 |
| 10 | 3 | 0.55 | 2 | 0.66 | 2 | 0.91 | 3 | 0.50 | 3 | 0.73 |
| 11 | 2 | 0.84 | 2 | 0.56 | 2 | 0.87 | 3 | 0.70 | 3 | 0.72 |
| 12 | 3 | 0.66 | 2 | 0.40 | 2 | 0.88 | 3 | 0.54 | 3 | 0.69 |
| 13 | 2 | 0.83 | 2 | 0.67 | 2 | 0.91 | 3 | 0.72 | 3 | 0.74 |
| 14 | 3 | 0.62 | 2 | 0.17 | 2 | 0.75 | 3 | 0.45 | 3 | 0.21 |
| 15 | 2 | 0.72 | 2 | 0.61 | 2 | 0.88 | 3 | 0.67 | 3 | 0.73 |
| 16 | 2 | 0.84 | 2 | 0.69 | 2 | 0.29 | 3 | 0.72 | 3 | 0.18 |
| 17 | 2 | 0.85 | 2 | 0.72 | 2 | 0.90 | 3 | 0.73 | 3 | 0.74 |
| 18 | 2 | 0.83 | 2 | 0.70 | 2 | 0.90 | 3 | 0.73 | 3 | 0.14 |
| 19 | 2 | 0.71 | 2 | 0.65 | 2 | 0.88 | 3 | 0.29 | 3 | 0.70 |
| 20 | 2 | 0.82 | 2 | 0.63 | 2 | 0.90 | 3 | 0.72 | 3 | 0.73 |
| 21 | 2 | 0.57 | 2 | 0.48 | 2 | 0.83 | 3 | 0.56 | 3 | 0.53 |
| 22 | 2 | 0.83 | 2 | 0.60 | 2 | 0.90 | 3 | 0.72 | 3 | 0.70 |

**Table 2. Number of significantly correlated pairs among all 231 pairs.**

| | Adjusted *P*-values | | | |
|---|---|---|---|---|
| | Max correlated set | | | |
| | >0.01 | <0.01 | >0.05 | <0.05 |
| CD4 T cells | 21 | 210 | 1 | 230 |
| Monocytes | 21 | 210 | 18 | 213 |
| Neutrophils | 9 | 222 | 4 | 227 |
| | Clinically correlated set | | | |
| | >0.01 | <0.01 | >0.05 | <0.05 |
| CD4 T cells | 61 | 170 | 58 | 173 |
| Neutrophils | 55 | 176 | 47 | 184 |

## Selection of genomic regions and variants, and their biological validation

Following the described procedures, genomic regions and variants were identified together with the included/associated genes for genomic regions and variants using biomaRt and SNPnexus, respectively. Transcription factor-binding sites (TFBSs) associated with genomic variants were also identified using SNPnexus. After collecting the genes identified for individual autosomes, Enrichr was used to identify TFs that targeted genes included in the

**Table 3. Correlation between clinical data and $u_{\ell_1 j}$.**

| | $\ell_1$ | 2 | 3 | 4 | 5 |
|---|---|---|---|---|---|
| | | CD4 T cells | | | |
| Glycoalbumin | Corr. | -0.10 | 0.07 | -0.02 | -0.08 |
| | P-value | 0.31 | 0.31 | 0.44 | 0.31 |
| Cystatin C | Corr. | -0.08 | 0.44 | -0.44 | -0.20 |
| | P-value | 0.31 | $4.05 \times 10^{-5}$ | $4.05 \times 10^{-5}$ | 0.16 |
| A blood-sugar | Corr. | -0.07 | 0.13 | -0.12 | -0.10 |
| | P-value | 0.32 | 0.27 | 0.31 | 0.31 |
| HbA1c | Corr. | -0.15 | 0.07 | -0.03 | -0.08 |
| | P-value | 0.22 | 0.31 | 0.44 | 0.31 |
| Number of red blood cells | Corr. | -0.167 | -0.22 | 0.15 | -0.06 |
| | P-value | 0.22 | 0.15 | 0.22 | 0.34 |
| Hemoglobin amount | Corr. | -0.08 | -0.06 | -0.00 | -0.16 |
| | P-value | 0.31 | 0.33 | 0.49 | 0.22 |
| Hematocrit value | Corr. | -0.08 | -0.09 | 0.01 | -0.13 |
| | P-value | 0.31 | 0.31 | 0.47 | 0.27 |
| | | Monocytes | | | |
| Glycoalbumin | Corr. | -0.07 | -0.02 | -0.04 | 0.10 |
| | P-value | 0.50 | 0.50 | 0.50 | 0.50 |
| Cystatin C | Corr. | -0.09 | -0.08 | -0.07 | 0.04 |
| | P-value | 0.50 | 0.50 | 0.50 | 0.50 |
| A blood-sugar | Corr. | 0.02 | 0.00 | 0.02 | 0.12 |
| | P-value | 0.50 | 0.50 | 0.50 | 0.50 |
| HbA1c | Corr. | -0.07 | 0.00 | -0.01 | 0.17 |
| | P-value | 0.50 | 0.50 | 0.50 | 0.50 |
| Number of red blood cells | Corr. | -0.02 | 0.01 | -0.01 | 0.02 |
| | P-value | 0.50 | 0.50 | 0.50 | 0.50 |
| Hemoglobin amount | Corr. | 0.12 | -0.03 | -0.06 | -0.05 |
| | P-value | 0.50 | 0.50 | 0.50 | 0.50 |
| Hematocrit value | Corr. | 0.12 | -0.01 | -0.03 | -0.05 |
| | P-value | 0.50 | 0.50 | 0.50 | 0.50 |
| | | Neutrophils | | | |
| Glycoalbumin | Corr. | 0.08 | -0.09 | 0.18 | 0.12 |
| | P-value | 0.32 | 0.32 | 0.21 | 0.27 |
| Cystatin C | Corr. | -0.11 | 0.21 | 0.04 | -0.16 |
| | P-value | 0.27 | 0.14 | 0.37 | 0.22 |
| A blood-sugar | Corr. | -0.06 | 0.38 | 0.13 | -0.25 |
| | P-value | 0.35 | $2.57 \times 10^{-3}$ | 0.27 | 0.08 |
| HbA1c | Corr. | -0.02 | 0.25 | 0.08 | -0.20 |
| | P-value | 0.43 | 0.08 | 0.32 | 0.14 |
| Number of red blood cells | Corr. | 0.09 | -0.03 | -0.14 | -0.05 |
| | P-value | 0.32 | 0.41 | 0.27 | 0.36 |
| Hemoglobin amount | Corr. | 0.05 | 0.07 | -0.12 | -0.11 |
| | P-value | 0.36 | 0.35 | 0.27 | 0.27 |
| Hematocrit value | Corr. | 0.05 | 0.08 | -0.11 | -0.17 |
| | P-value | 0.36 | 0.32 | 0.27 | 0.21 |

**Table 4. TFs for genes identified by gene expression in "ENCODE and ChEA Consensus TFs from ChIP-X".**

| | Max correlated set |
|---|---|
| CD4 T cells | RUNX1, SPI1, RELA, TCF3, NELFE, CEBPD, BCL3, PML, ZMIZ1, SRF, GATA1, GATA2, IRF8, IRF1, MYC, ATF2, TAF7, CHD1, NFE2L2, KLF4, ERG, STAT3, PBX3, KAT2A, FOXA1, CEBPB |
| Neutrophils | SPI1, RUNX1, TCF3, CEBPD, PML, NELFE, RELA, BCL3, SRF, GATA1, CREB1, GATA2, ZMIZ1, BRCA1, ATF2, PPARG, KLF4 |
| Monocytes | CEBPD, MYC, NELFE, TAF1, ZMIZ1, SRF, MAX, BRCA1, CEBPB, RELA, BCL3, SPI1, TCF3, BCLAF1, ZBTB33, KLF4, CREB1, ELF1, FLI1, CHD1, NFIC, SPI1, YY1, STAT3 |
| | Clinically correlated set |
| CD4 T cells | TAF7, KAT2A, NELFE, PML, MYC, ATF2, TAF1, ZMIZ1, TCF3, CEBPD, SPI1, RELA, CREB1, RUNX1, BCL3, MAX, BRCA1, SRF, KLF4, CHD1, FLI1, STAT3, PBX3, ZBTB33, BCLAF1, RFX5, NFIC, E2F1, CEBPB, EGR1, UBTF, YY1, ELF1, GABPA, NFYA, ZC3H11A |
| Neutrophils | RUNX1, SPI1, RELA, TCF3, NELFE, CEBPD, BCL3, PML, ZMIZ1, SRF, GATA1, GATA2, IRF8, IRF1, MYC, ATF2, TAF7, CHD1, NFE2L2, KLF4, ERG, STAT3, PBX3, KAT2A, FOXA1, CEBPB |

**Table 5. KEGG Human 2019 (for TFs listed in the "CD4 T cells" category under the "Max correlated set" of Table 4).**

| Term | Overlap | P-value | Adjusted P-value |
|---|---|---|---|
| Pathways in cancer | 22/530 | $4.47 \times 10^{-25}$ | $5.54 \times 10^{-23}$ |
| Transcriptional misregulation in cancer | 14/186 | $2.71 \times 10^{-19}$ | $1.68 \times 10^{-17}$ |
| Hepatitis B | 11/163 | $1.03 \times 10^{-14}$ | $4.26 \times 10^{-13}$ |
| Epstein-Barr virus infection | 11/201 | $1.05 \times 10^{-13}$ | $3.26 \times 10^{-12}$ |
| Breast cancer | 10/147 | $1.79 \times 10^{-13}$ | $4.43 \times 10^{-12}$ |
| Acute myeloid leukemia | 8/66 | $5.14 \times 10^{-13}$ | $1.06 \times 10^{-11}$ |
| Kaposi sarcoma-associated herpesvirus infection | 10/186 | $1.91 \times 10^{-12}$ | $3.38 \times 10^{-11}$ |
| Viral carcinogenesis | 10/201 | $4.14 \times 10^{-12}$ | $6.42 \times 10^{-11}$ |
| Th1 and Th2 cell differentiation | 8/92 | $8.04 \times 10^{-12}$ | $1.11 \times 10^{-10}$ |
| Th17 cell differentiation | 8/107 | $2.76 \times 10^{-11}$ | $3.42 \times 10^{-10}$ |

**Table 6. TFs for genes identified by DNA methylation in "ChEA 2016".**

| | Max correlated set |
|---|---|
| CD4 T cells | SMC4, LXR, EGR1, P68, ERG, KDM2B, GATA2, BCL6, DACH1, CTCF, BCOR, SMAD2/3, OCT4, SCL, ELK3, KLF4, VDR, TFAP2C, DROSHA, MAF, CREB1, MYCN, P300, TP63, CTCF |
| Neutrophils | EGR1, LXR, SMC4, KDM2B, P68, BCL6, GATA2, DACH1, VDR, BCOR, EZH2, KLF4, DROSHA, ERG, OCT4, CTCF, SCL, TP63, ELK3, TFAP2C, SMAD2/3, CREB1, E2A, MITF, SA1, P300, RACK7, MYCN, CTNNB1, KDM5A, SOX9, RUNX1, MAF |
| Monocytes | EGR1, LXR, SMC4, KDM2B, P68, SCL, BCL6, GATA2, KLF4, BCOR, TP63, ERG, VDR, DROSHA, EZH2, DACH1, OCT4, CTCF, TFAP2C, ELK3, P300, SMAD2/3, E2A, MITF, RACK7, RUNX1, CREB1, MYCN, SA1, MAF |
| | Clinically correlated set |
| CD4 T cells | JARID2, EZH2, PHC1, TP53, BMI, EED, SUZ12, CBX2, RNF2, MTF2, RING1B, POU5F1, KLF4, STAT3, ERG, KDM5B, TP63, SMAD3 |
| Neutrophils | EGR1, LXR, SMC4, KDM2B, P68, BCL6, GATA2, DACH1, VDR, EZH2, KLF4, DROSHA, OCT4, BCOR, TP63, SCL, ELK3, TFAP2C, ERG, CTCF, SMAD2/3, RACK7, MITF, CREB1, SA1, MYCN, E2A, P300, CTNNB1, MAF |

**Table 7. KEGG Human 2019 (for TFs listed in the "CD4 T cells" category under the "Max correlated set" of Table 6).**

| Term | Overlap | P-value | Adjusted P-value |
|---|---|---|---|
| Inflammatory bowel disease (IBD) | 3/65 | $7.16 \times 10^{-5}$ | $2.67 \times 10^{-3}$ |
| Transcriptional misregulation in cancer | 4/186 | $7.86 \times 10^{-5}$ | $2.67 \times 10^{-3}$ |
| Human T-cell leukemia virus 1 infection | 4/219 | $1.48 \times 10^{-4}$ | $3.35 \times 10^{-3}$ |
| AGE-RAGE signaling pathway in diabetic complications | 3/100 | $2.58 \times 10^{-4}$ | $4.16 \times 10^{-3}$ |
| Parathyroid hormone synthesis, secretion and action | 3/106 | $3.06 \times 10^{-4}$ | $4.16 \times 10^{-3}$ |
| Relaxin signaling pathway | 3/130 | $5.56 \times 10^{-4}$ | $5.10 \times 10^{-3}$ |
| FoxO signaling pathway | 3/132 | $5.81 \times 10^{-4}$ | $5.10 \times 10^{-3}$ |
| Apelin signaling pathway | 3/137 | $6.48 \times 10^{-4}$ | $5.10 \times 10^{-3}$ |
| Signaling pathways regulating pluripotency of stem cells | 3/139 | $6.75 \times 10^{-4}$ | $5.10 \times 10^{-3}$ |
| Hepatitis B | 3/163 | $1.07 \times 10^{-3}$ | $7.29 \times 10^{-3}$ |

**Table 8. KEGG Human 2019 (for TFs listed in the "Neutrophils" category under the "Max correlated set" of Table 4).**

| Term | Overlap | P-value | Adjusted P-value |
|---|---|---|---|
| Transcriptional misregulation in cancer | 6/186 | $6.78 \times 10^{-9}$ | $6.51 \times 10^{-7}$ |
| Human T-cell leukemia virus 1 infection | 6/219 | $1.80 \times 10^{-8}$ | $8.65 \times 10^{-7}$ |
| Acute myeloid leukemia | 4/66 | $2.49 \times 10^{-7}$ | $7.97 \times 10^{-6}$ |
| Longevity regulating pathway | 4/102 | $1.44 \times 10^{-6}$ | $3.46 \times 10^{-5}$ |
| TNF signaling pathway | 4/110 | $1.95 \times 10^{-6}$ | $3.75 \times 10^{-5}$ |
| Osteoclast differentiation | 4/127 | $3.46 \times 10^{-6}$ | $5.54 \times 10^{-5}$ |
| Cocaine addiction | 3/49 | $9.17 \times 10^{-6}$ | $1.26 \times 10^{-4}$ |
| Viral carcinogenesis | 4/201 | $2.13 \times 10^{-5}$ | $2.55 \times 10^{-4}$ |
| Pathways in cancer | 5/530 | $6.09 \times 10^{-5}$ | $6.50 \times 10^{-4}$ |
| Relaxin signaling pathway | 3/130 | $1.71 \times 10^{-4}$ | $1.64 \times 10^{-3}$ |

**Table 9. KEGG Human 2019 (for TFs listed in the "Neutrophils" category under the "Max correlated set" of Table 6).**

| Term | Overlap | P-value | Adjusted P-value |
|---|---|---|---|
| Transcriptional misregulation in cancer | 6/186 | $6.46^{-7}$ | $3.88^{-5}$ |
| Signaling pathways regulating pluripotency of stem cells | 4/139 | $8.81^{-5}$ | $2.64^{-3}$ |
| Inflammatory bowel disease (IBD) | 3/65 | $1.82^{-4}$ | $3.65^{-3}$ |
| Adherens junction | 3/72 | $2.47^{-4}$ | $3.71^{-3}$ |
| Colorectal cancer | 3/86 | $4.17^{-4}$ | $5.01^{-3}$ |
| AGE-RAGE signaling pathway in diabetic complications | 3/100 | $6.48^{-4}$ | $6.48^{-3}$ |
| Th17 cell differentiation | 3/107 | $7.90^{-4}$ | $6.77^{-3}$ |
| FoxO signaling pathway | 3/132 | $1.45^{-3}$ | $1.07^{-2}$ |
| Apelin signaling pathway | 3/137 | $1.61^{-3}$ | $1.07^{-2}$ |
| Pathways in cancer | 5/530 | $1.89^{-3}$ | $1.12^{-2}$ |

genomic regions and validate their biological significance by applying KTD-based unsupervised FE.

**"Max correlated set": CD4 T cells.** We identified 221 and 536 genomic regions for gene expression and DNA methylation, respectively, as well as 1,174,607 genomic variants that were supposed to coincide with the subject profiles represented by $u_{\ell_1 j}$ listed in the column "CD4 T Cells" under the "Max correlated set" in Table 1. A total of 419 and 590 genes were

**Table 10. KEGG Human 2019 (for TFs listed in "Monocytes" category under "Max correlated set" of Table 4).**

| Term | Overlap | P-value | Adjusted P-value |
|---|---|---|---|
| Transcriptional misregulation in cancer | 7/186 | $1.16 \times 10^{-9}$ | $1.29 \times 10^{-7}$ |
| Human T-cell leukemia virus 1 infection | 6/219 | $1.39 \times 10^{-7}$ | $7.72 \times 10^{-6}$ |
| Acute myeloid leukemia | 4/66 | $9.13 \times 10^{-7}$ | $3.38 \times 10^{-5}$ |
| TNF signaling pathway | 4/110 | $7.08 \times 10^{-6}$ | $1.96 \times 10^{-4}$ |
| Signaling pathways regulating pluripotency of stem cells | 4/139 | $1.79 \times 10^{-5}$ | $3.96 \times 10^{-4}$ |
| Hepatitis B | 4/163 | $3.34 \times 10^{-5}$ | $6.17 \times 10^{-4}$ |
| Kaposi sarcoma-associated herpesvirus infection | 4/186 | $5.58 \times 10^{-5}$ | $8.86 \times 10^{-4}$ |
| Viral carcinogenesis | 4/201 | $7.55 \times 10^{-5}$ | $1.05 \times 10^{-3}$ |
| Human cytomegalovirus infection | 4/225 | $1.17 \times 10^{-4}$ | $1.44 \times 10^{-3}$ |
| Small cell lung cancer | 3/93 | $1.61 \times 10^{-4}$ | $1.79 \times 10^{-3}$ |

**Table 11. KEGG Human 2019 (for TFs listed in the "Monocytes" category under the "Max correlated set" of Table 6).**

| Term | Overlap | P-value | Adjusted P-value |
|---|---|---|---|
| Transcriptional misregulation in cancer | 6/186 | $3.62 \times 10^{-7}$ | $2.64 \times 10^{-5}$ |
| Inflammatory bowel disease (IBD) | 3/65 | $1.38 \times 10^{-4}$ | $5.04 \times 10^{-3}$ |
| Human T-cell leukemia virus 1 infection | 4/219 | $3.49 \times 10^{-4}$ | $7.30 \times 10^{-3}$ |
| AGE-RAGE signaling pathway in diabetic complications | 3/100 | $4.92 \times 10^{-4}$ | $7.30 \times 10^{-3}$ |
| Parathyroid hormone synthesis, secretion and action | 3/106 | $5.84 \times 10^{-4}$ | $7.30 \times 10^{-3}$ |
| Th17 cell differentiation | 3/107 | $6.00 \times 10^{-4}$ | $7.30 \times 10^{-3}$ |
| Relaxin signaling pathway | 3/130 | $1.06 \times 10^{-3}$ | $9.35 \times 10^{-3}$ |
| FoxO signaling pathway | 3/132 | $1.10 \times 10^{-3}$ | $9.35 \times 10^{-3}$ |
| Apelin signaling pathway | 3/137 | $1.23 \times 10^{-3}$ | $9.35 \times 10^{-3}$ |
| Signaling pathways regulating pluripotency of stem cells | 3/139 | $1.28 \times 10^{-3}$ | $9.35 \times 10^{-3}$ |

**Table 12. KEGG Human 2019 (for TFs listed in the "CD4 T cells" category under the "Clinically correlated set" of Table 4).**

| Term | Overlap | P-value | Adjusted P-value |
|---|---|---|---|
| Transcriptional misregulation in cancer | 10/186 | $7.82 \times 10^{-13}$ | $9.38 \times 10^{-11}$ |
| Human T-cell leukemia virus 1 infection | 10/219 | $4.00 \times 10^{-12}$ | $2.40 \times 10^{-10}$ |
| Acute myeloid leukemia | 6/66 | $1.84 \times 10^{-9}$ | $7.38 \times 10^{-8}$ |
| Hepatitis B | 6/163 | $4.25 \times 10^{-7}$ | $1.28 \times 10^{-5}$ |
| Viral carcinogenesis | 6/201 | $1.45 \times 10^{-6}$ | $3.02 \times 10^{-5}$ |
| TNF signaling pathway | 5/110 | $1.51 \times 10^{-6}$ | $3.02 \times 10^{-5}$ |
| Human cytomegalovirus infection | 6/225 | $2.79 \times 10^{-6}$ | $4.78 \times 10^{-5}$ |
| Pathways in cancer | 8/530 | $3.63 \times 10^{-6}$ | $5.44 \times 10^{-5}$ |
| Chronic myeloid leukemia | 4/76 | $1.03 \times 10^{-5}$ | $1.38 \times 10^{-4}$ |
| Tuberculosis | 5/179 | $1.63 \times 10^{-5}$ | $1.96 \times 10^{-4}$ |

included in the genomic regions selected for gene expression and DNA methylation, respectively. A total of 14,346 genes were associated with genomic variants. By uploading 419 genes to Enrichr for gene expression analysis, 26 TFs with threshold-adjusted P-values less than 0.05 were identified in the "ChEA & ENCODE consensus" category (Table 4). To validate their biological significance, these 26 TFs were uploaded to Enrichr and found to form a biologically significant set (Table 5). Furthermore, 25 TFs with threshold-adjusted P-values less

**Table 13. KEGG Human 2019 (for TFs listed in the "CD4 T cells" category under the "Clinically correlated set" of Table 6).**

| Term | Overlap | P-value | Adjusted P-value |
|---|---|---|---|
| Signaling pathways regulating pluripotency of stem cells | 5/139 | $1.20 \times 10^{-7}$ | $8.17 \times 10^{-6}$ |
| Pancreatic cancer | 3/75 | $3.97 \times 10^{-5}$ | $1.35 \times 10^{-3}$ |
| MicroRNAs in cancer | 4/299 | $1.27 \times 10^{-4}$ | $2.88 \times 10^{-3}$ |
| Hepatitis B | 3/163 | $3.96 \times 10^{-4}$ | $6.74 \times 10^{-3}$ |
| Inflammatory bowel disease (IBD) | 2/65 | $1.54 \times 10^{-3}$ | $1.80 \times 10^{-2}$ |
| Non-small cell lung cancer | 2/66 | $1.59 \times 10^{-3}$ | $1.80 \times 10^{-2}$ |
| Chronic myeloid leukemia | 2/76 | $2.10 \times 10^{-3}$ | $2.04 \times 10^{-2}$ |
| Colorectal cancer | 2/86 | $2.67 \times 10^{-3}$ | $2.27 \times 10^{-2}$ |
| Prostate cancer | 2/97 | $3.39 \times 10^{-3}$ | $2.44 \times 10^{-2}$ |
| AGE-RAGE signaling pathway in diabetic complications | 2/100 | $3.59 \times 10^{-3}$ | $2.44 \times 10^{-2}$ |

**Table 14. KEGG Human 2019 (for TFs listed in the "Neutrophils" category under the "Clinically correlated set" of Table 4).**

| Term | Overlap | P-value | Adjusted P-value |
|---|---|---|---|
| Transcriptional misregulation in cancer | 9/186 | $1.17 \times 10^{-12}$ | $1.21 \times 10^{-10}$ |
| Acute myeloid leukemia | 6/66 | $2.24 \times 10^{-10}$ | $1.16 \times 10^{-8}$ |
| Human T-cell leukemia virus 1 infection | 7/219 | $9.45 \times 10^{-9}$ | $3.28 \times 10^{-7}$ |
| TNF signaling pathway | 5/110 | $2.75 \times 10^{-7}$ | $7.16 \times 10^{-6}$ |
| Pathways in cancer | 7/530 | $3.75 \times 10^{-6}$ | $7.79 \times 10^{-5}$ |
| Viral carcinogenesis | 5/201 | $5.40 \times 10^{-6}$ | $9.37 \times 10^{-5}$ |
| Signaling pathways regulating pluripotency of stem cells | 4/139 | $2.97 \times 10^{-5}$ | $4.41 \times 10^{-4}$ |
| Hepatitis B | 4/163 | $5.53 \times 10^{-5}$ | $7.18 \times 10^{-4}$ |
| Prolactin signaling pathway | 3/70 | $1.01 \times 10^{-4}$ | $1.16 \times 10^{-3}$ |
| Chronic myeloid leukemia | 3/76 | $1.29 \times 10^{-4}$ | $1.22 \times 10^{-3}$ |

**Table 15. KEGG Human 2019 (for TFs listed in "CD4 T cells" category under the "Clinically correlated set" of Table 6).**

| Term | Overlap | P-value | Adjusted P-value |
|---|---|---|---|
| Transcriptional misregulation in cancer | 5/186 | $9.20 \times 10^{-6}$ | $7.73 \times 10^{-4}$ |
| Signaling pathways regulating the pluripotency of stem cells | 4/139 | $6.08 \times 10^{-5}$ | $2.55 \times 10^{-3}$ |
| Inflammatory bowel disease (IBD) | 3/65 | $1.38 \times 10^{-4}$ | $3.86 \times 10^{-3}$ |
| Adherens junction | 3/72 | $1.87 \times 10^{-4}$ | $3.93 \times 10^{-3}$ |
| Colorectal cancer | 3/86 | $3.16 \times 10^{-4}$ | $4.73 \times 10^{-3}$ |
| Human T-cell leukemia virus 1 infection | 4/219 | $3.49 \times 10^{-4}$ | $4.73 \times 10^{-3}$ |
| Prostate cancer | 3/97 | $4.50 \times 10^{-4}$ | $4.73 \times 10^{-3}$ |
| AGE-RAGE signaling pathway in diabetic complications | 3/100 | $4.92^{-4}$ | $4.73^{-3}$ |
| Melanogenesis | 3/101 | $5.07^{-4}$ | $4.73^{-3}$ |
| Parathyroid hormone synthesis, secretion and action | 3/106 | $5.84^{-4}$ | $4.90^{-3}$ |

than 0.05 were identified in the "ChEA 2016" category by uploading 590 genes for DNA methylation to Enrichr (Table 6). These 25 TFs formed a biologically significant set (Table 7).

**"Max correlated set": Neutrophils.** We identified 356 and 154 genomic regions for gene expression and DNA methylation, respectively, and 778,698 genomic variants supposed to

be coincident with the subject profiles represented by $u_{\ell_1 j}$ listed in the column "Neutrophils" under "Max correlated set" in Table 1. A total of 490 and 500 genes were included in the genomic regions selected for gene expression and DNA methylation, respectively. Furthermore, 15,356 genes were associated with genomic variants. Of the 490 genes involved in gene expression uploaded to Enrichr, 17 TFs with threshold-adjusted *P*-values less than 0.05 were identified in the "ChEA & ENCODE consensus" category (Table 4). These 17 TFs formed a biologically significant set (Table 8). Furthermore, by uploading 500 genes identified for DNA methylation, 33 TFs with threshold-adjusted *P*-values less than 0.05 were identified in the "ChEA 2016" category (Table 6). These 33 TFs formed a biologically significant set (Table 9).

**"Max correlated set": Monocytes.** We identified 182 and 558 genomic regions for gene expression and DNA methylation, respectively, as well as 1,105,748 genomic variants that were supposed to coincide with the subject profiles represented by $u_{\ell_1 j}$ in the column "Monocytes" under the "Max correlated set" in Table 1. In total, 453 and 1,015 genes were included in the genomic regions selected for gene expression and DNA methylation, respectively.

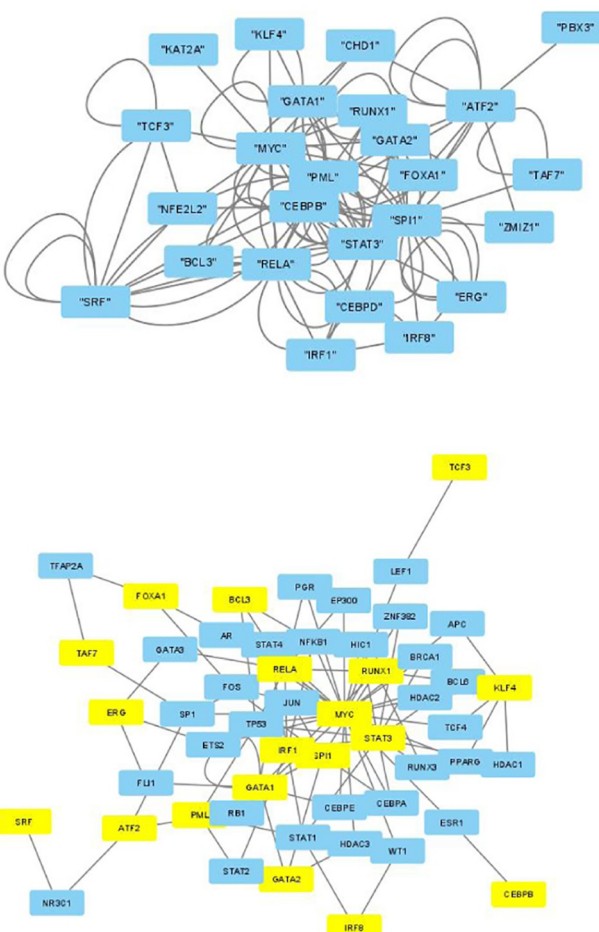

**Fig 3. Regulatory network between TFs in the "CD4 T cells category" under the "Max correlated set" in Table 4.** Upper: Regnetwork web, lower: TTRUST2. Blue genes in Regnetwork web and yellow genes in TTRUST2 are TFs in Table 4. Blue genes in TTRUST2 are associated with these.

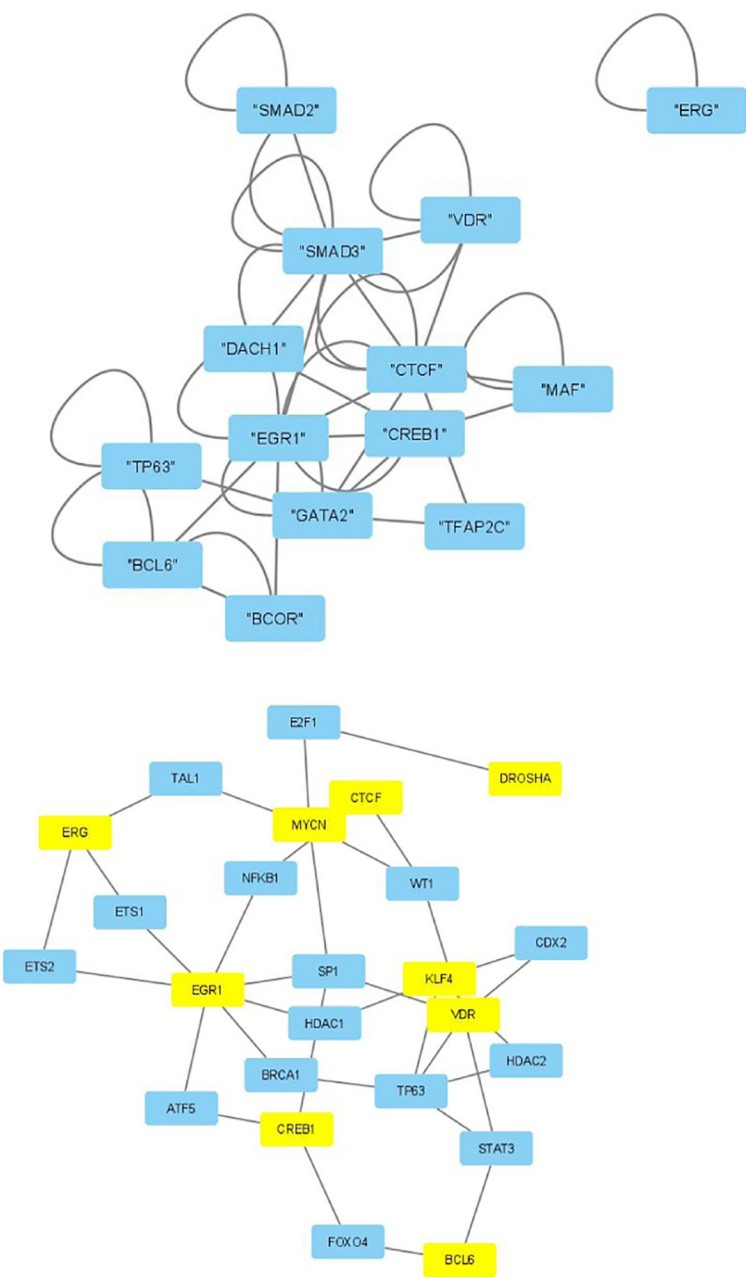

**Fig 4. Regulatory network between TFs in the "CD4 T cells category" under the "Max correlated set" in Table 6.** Upper: Regnetwork web, lower: TTRUST2. Blue genes in Regnetwork web and yellow genes in TTRUST2 are TFs in Table 6. Blue genes in TTRUST2 are associated with these.

Furthermore, 14,032 genes were associated with genomic variants. Twenty-four TFs with threshold-adjusted *P*-values less than 0.05 were identified in the "ChEA & ENCODE consensus" category by uploading 182 genes identified for gene expression (Table 4). These 24 TFs formed a biologically significant set (Table 10). Furthermore, 30 TFs with threshold-adjusted *P*-values less than 0.05 were identified in the "ChEA 2016" category by uploading 558 genes

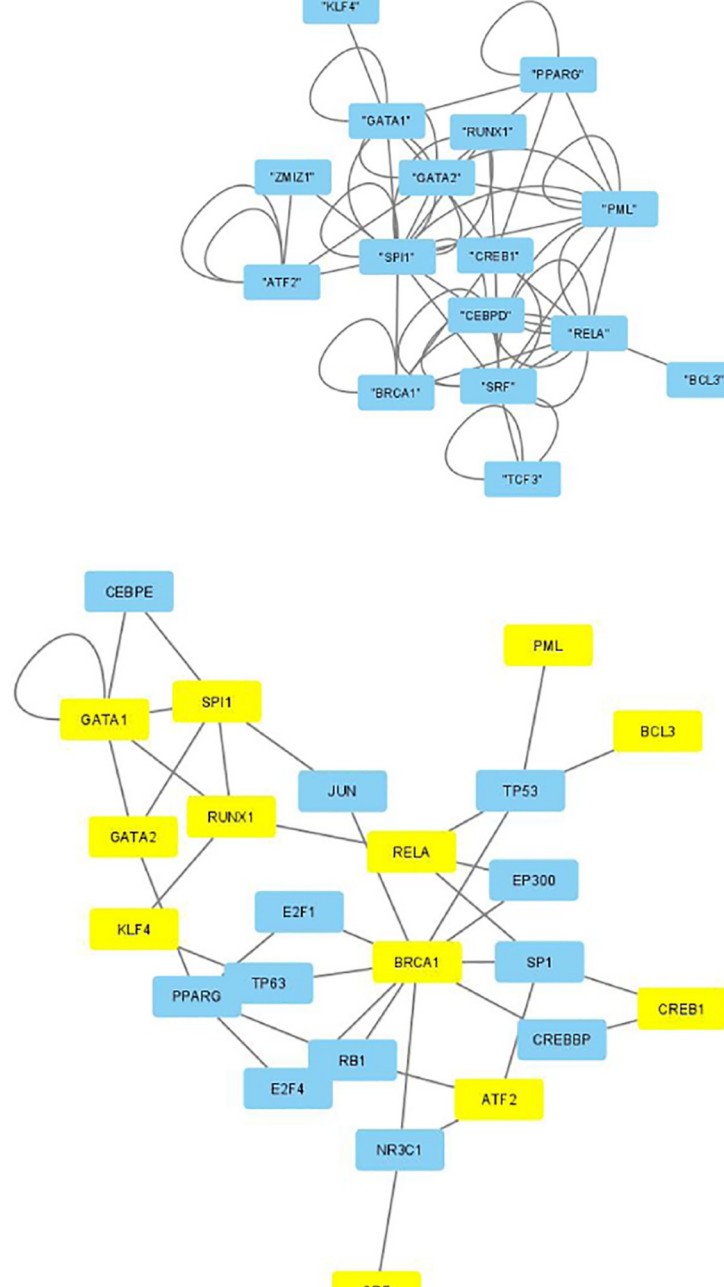

**Fig 5. Regulatory network between TFs in "Neutrophils" under the "Max correlated set" in Table 4.** Upper: Regnetwork web, lower: TTRUST2. Blue genes in Regnetwork web and yellow genes in TTRUST2 are TFs in Table 4. Blue genes in TTRUST2 are associated with these.

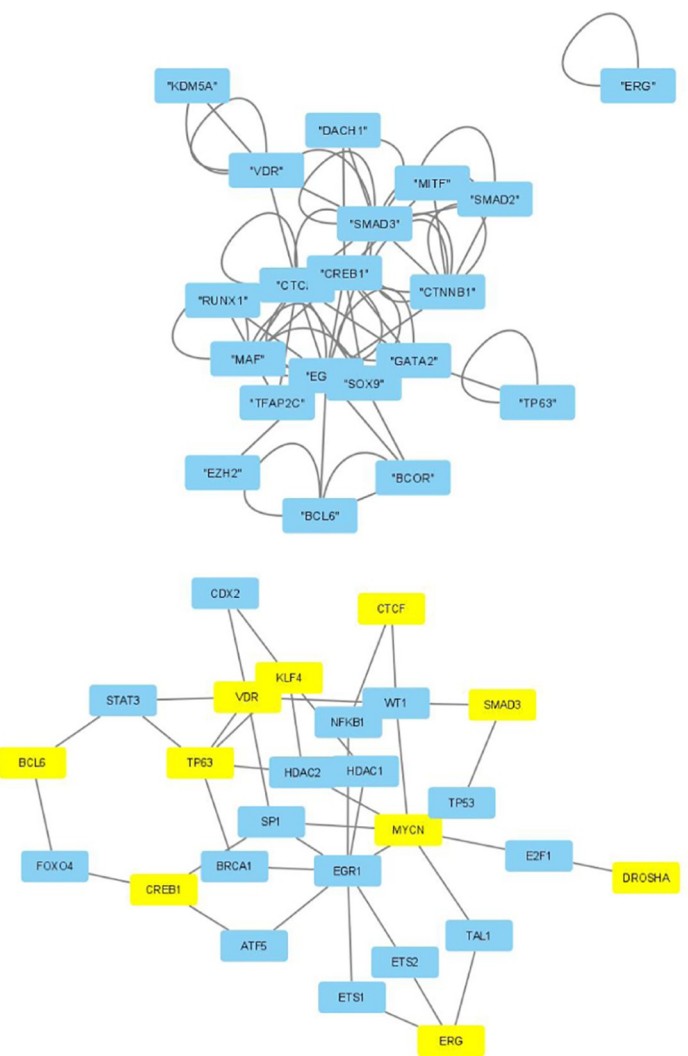

**Fig 6. Regulatory network between TFs in "Neutrophils" under the "Max correlated set" in Table 6.** Upper: Regnetwork web, lower: TTRUST2. Blue genes in Regnetwork web and yellow genes in TTRUST2 are TFs in Table 6. Blue genes in TTRUST2 are associated with these.

for DNA methylation (Table 6). These 30 TFs formed a biologically significant set (Table 11).

**"Clinically correlated set": CD4 T cells.** We identified 425 and 281 genomic regions for gene expression and DNA methylation, respectively as well as 1,073,649 genomic variants that are supposed to coincide with the subject profiles represented by $u_{\ell_1 j}$ in the column "CD4 T cell" under "Clinically correlated set" in Table 1. In total, 794 and 412 genes were included in the genomic regions selected for gene expression and DNA methylation, respectively. Furthermore, 13,178 genes were associated with genomic variants. After uploading 794 genes for gene expression, 36 TFs with threshold-adjusted *P*-values less than 0.05 were identified in the "ChEA & ENCODE consensus" category (Table 4). These 36 TFs formed a biologically significant set (Table 12). Furthermore, 18 TFs with threshold-adjusted *P*-values less than 0.05 were identified in the "ChEA 2016" category by uploading 412 genes for DNA

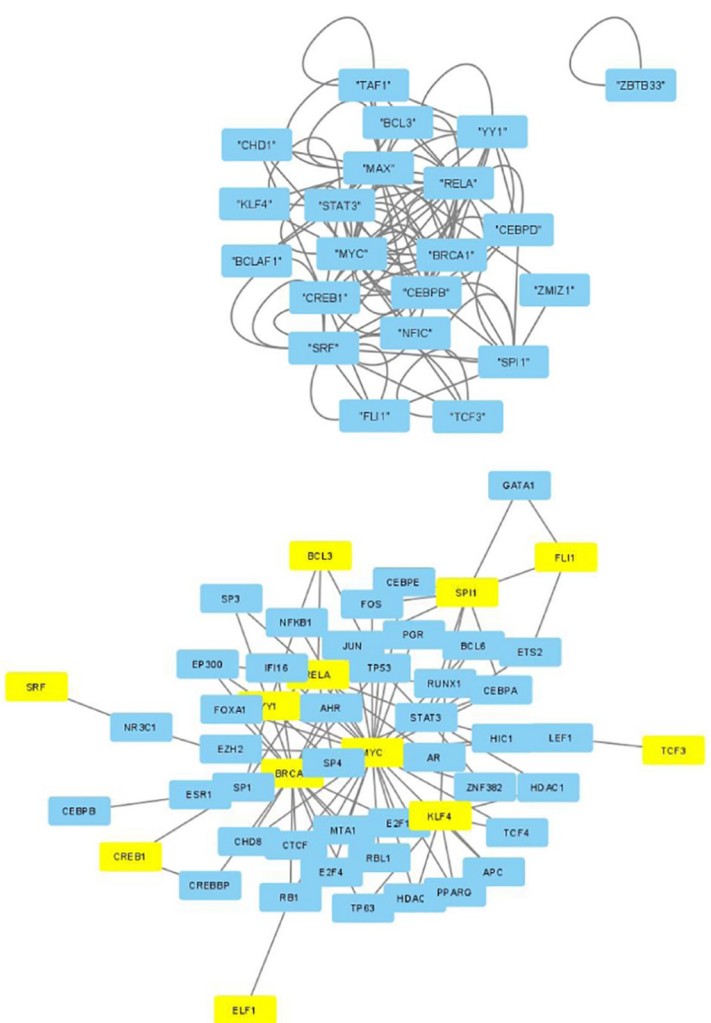

**Fig 7. Regulatory network between TFs in "Monocytes" under the "Max correlated set" in Table 4.** Upper: Regnetworkweb, lower: TTRUST2. Blue genes in Regnetworkweb and yellow genes in TTRUST2 are TFs in Table 4. Blue genes in TTRUST2 are associated with these.

methylation to Enrichr, (Table 6). These 18 TFs formed a biologically significant set (Table 13).

**"Clinically correlated set": Neutrophils.** We identified 380 and 541 genomic regions for gene expression and DNA methylation, respectively, as well as 63,894 genomic variants that are supposed to coincide with subject profiles represented by $u_{\ell_1 j}$ in the column "Neutrophils" under the "Clinically correlated set" in Table 1. A total of 610 and 499 genes were included in the genomic regions selected for gene expression and DNA methylation, respectively. Furthermore, 3,292 genes were associated with genomic variants. By uploading the 610 genes identified for gene expression to Enrichr, 26 TFs with threshold-adjusted *P*-values less than 0.05 were identified in the "ChEA & ENCODE consensus" category (Table 4). These 26 TFs formed a biologically significant set (Table 14). Furthermore, by uploading 499 genes for DNA methylation, 30 TFs with threshold adjusted *P*-values less than 0.05 were

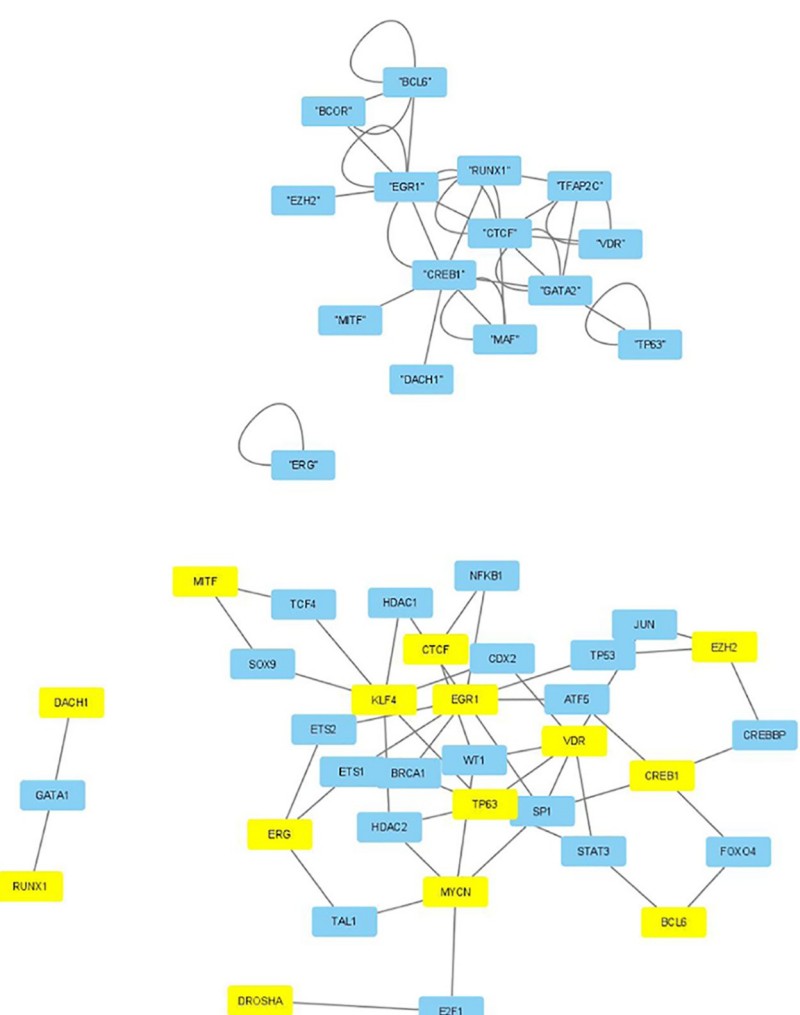

**Fig 8. Regulatory network between TFs in "Monocytes" under the "Max correlated set" in Table 6.** Upper: Regnetworkweb, lower: TTRUST2. Blue genes in Regnetworkweb and yellow genes in TTRUST2 are TFs in Table 6. Blue genes in TTRUST2 are associated with these.

identified in the "ChEA 2016" category (Table 6). These 30 TFs formed a biologically significant set (Table 15).

## Discussion

The selected genes were targeted by various TFs enriched in KEGG pathways; thus, genes with profiles coincident with the patient profiles expressed by the selected singular value vectors were biologically valid. The selected KEGG pathways were likely to express the biological properties of the participants' blood cells. If they are intraregulated, the identified genes and variants are probably effective. To determine whether the identified genes were intraregulated, we uploaded the selected TFs to two databases that validated the regulatory relationships between TFs: Regnetworkweb and TRRUST2. Regnetworkweb considers only direct regulatory relationships between TFs whereas TRRUST2 considers indirect regulatory relationships; for example, two TFs targeting the same genes (Figs 3–12). These are clearly

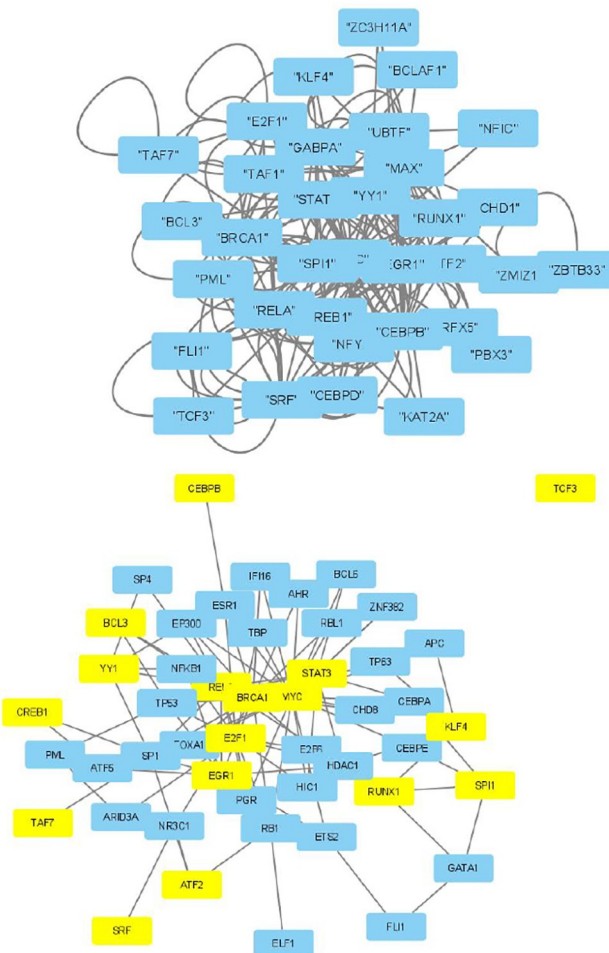

**Fig 9. Regulatory network between TFs in "CD4 T cells" under the "Clinically correlated set" in Table 4.** Upper: Regnetworkweb, lower: TTRUST2. Blue genes in Regnetwork web and yellow genes in TTRUST2 are TFs in Table 4. Blue genes in TTRUST2 are associated with these.

highly intracorrelated. Thus, in terms of regulatory relationships, the identified TFs are reasonable.

To validate the overlap between TFs that target genes identified based on gene expression or DNA methylation and those identified based on TFBS, the total number of human TFs must be determined, and we assume that it is approximately 2000 [27]. Table 16 shows the results of Fisher's test. TFs identified for gene expression significantly overlapped with TFBS associated with genomic variants, even though no significant overlaps were found between the TFBS identified using genomic variants and TFs identified for methylation (not shown here).

One might wonder why we did not compare our performance with that of existing methods. To the best of our knowledge, no other methods are comparable to ours. First, our analysis of association studies between gene expression, methylation, and genomic variants is free from location restrictions; this method can detect any kind of association between these genes, independent of their location along the genome. For example, we can identify interactions between genes and genomic variants that are distant from each other. This is

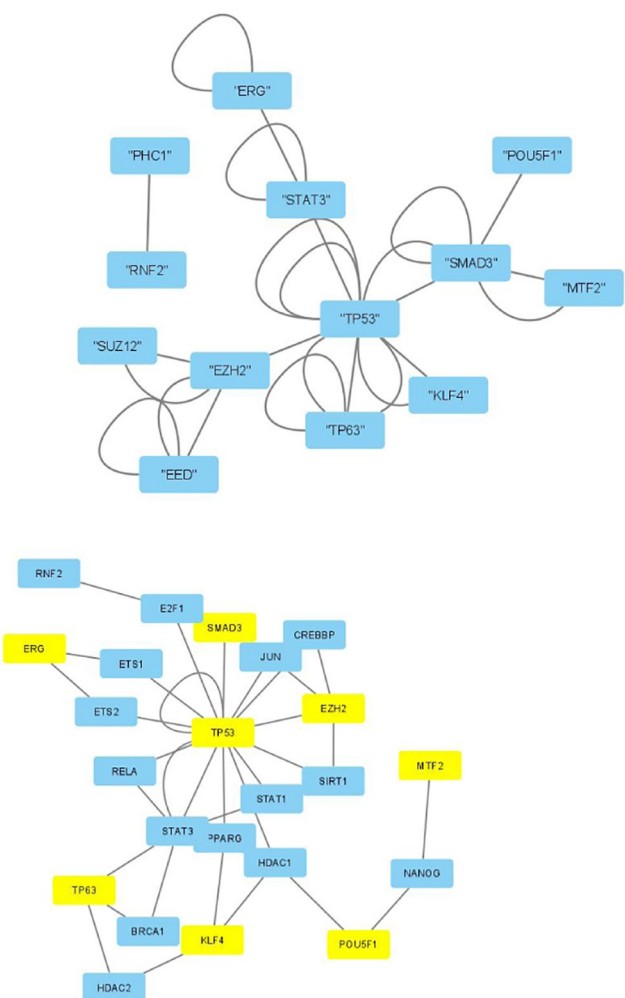

**Fig 10. Regulatory network between TFs in "CD4 T cells" under the "Clinically correlated set" in Table 6.** Upper: Regnetwork web, lower: TTRUST2. Blue genes in Regnetwork web and yellow genes in TTRUST2 are TFs in Table 6. Blue genes in TTRUST2 are associated with these.

because we can derive the singular value vectors attributed to the subjects, $u_{\ell_1 j}$, at the very beginning of the data analysis flow just after applying TD to $x_{jj'k}$. Genomic regions and/or variants were then selected based on singular value vectors $u_{\ell_1 i_k}$ attributed to genomic regions or variants $i_k$. Application of TD to $x_{jj'k}$ requires a very small amount of computational resources, as $x_{jj'k} \in \mathbb{R}^{M \times M \times K}$. To our knowledge, no other method can select genomic regions and variants using such a small amount of computational resources. In particular, treating genomic variants is difficult. For gene expression and methylation, these values can be averaged within individual genomic regions, resulting in a reduced dimension of $i_k$, that is, $N_k$. Nevertheless, this cannot be performed for genomic variants because the integers (1, 2, and 3) derived from the genomic variants are arbitrary. Averaging distinct integer numbers attributed to individual genomic variants can destroy the meaning of these integers. Despite this, our method is independent of the size of $N_k$, and can be applied to genomic variants as is. To the best of our knowledge, no other methods can perform this

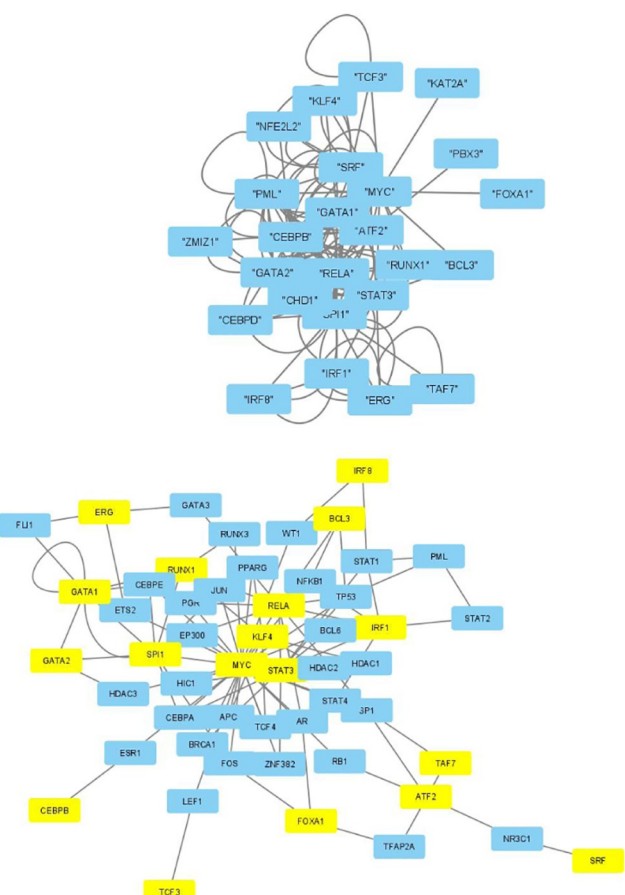

**Fig 11. Regulatory network between TFs in "Neutrophils" under the "Clinically correlated set" in Table 4.** Upper: Regnetwork web, lower: TTRUST2. Blue genes in Regnetwork web and yellow genes in TTRUST2 are TFs in Table 4. Blue genes in TTRUST2 are associated with these.

task; thus, we could not compare the performance of our method with that of any other method.

Our methods do not require prior knowledge of the subjects. Singular value vectors attributed to the subjects, $u_{\ell_1 j}$, can be generated by applying TD to $x_{jj'M}$, which does not require any additional information about the subjects. The selection of $u_{\ell_1 j}$ used to select $i_K$ was based on the coincidence between those computed for individual autosomes. Therefore, genomic regions and variants can be selected in a fully unsupervised manner. However, the selected genes were significantly targeted by multiple TFs that were enriched in KEGG pathway terms.

Several biological insights were obtained from this population-based study. One possible application to clinical studies is to compare the outcomes of the present study with those of other clinical studies. Generally, both population- and clinical-based studies have their own biases, and by comparing their outcomes with each other, we can validate their outcomes, which is impossible when only individual outcomes are present.

However, this method had several limitations. First, it is applicable to multiomics datasets that share samples. In addition, because this is an unsupervised method, if there are no significant results in the downward analyses, we have no way to improve the results.

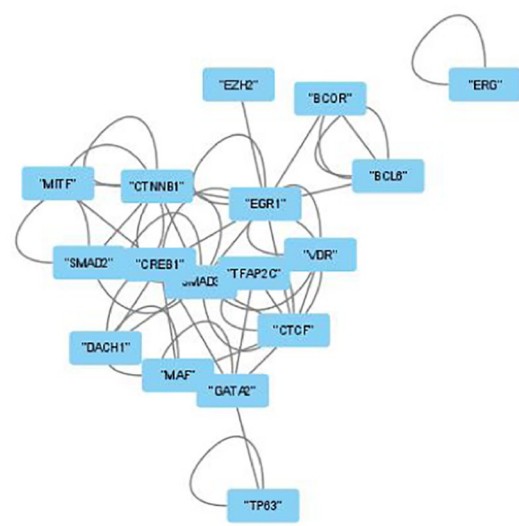

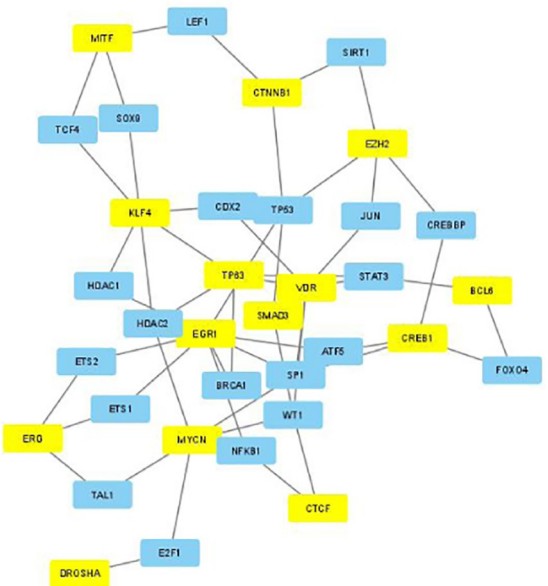

**Fig 12. Regulatory network between TFs in "Neutrophils" under the "Clinically correlated set" in Table 6.** Upper: Regnetwork web, lower: TTRUST2. Blue genes in Regnetwork web and yellow genes in TTRUST2 are TFs in Table 6. Blue genes in TTRUST2 are associated with these.

**Table 16. Fisher's exact tests between TFs in Table 4 and those identified through the TFBSs of genomic variants detected by KTD based unsupervised FE.**

| | | Gene expression | |
|---|---|---|---|
| Max correlated set | | | |
| CD4 T cells | | not selected | selected |
| TFBS | not selected | 1737 | 19 |
| | selected | 237 | 7 |
| odds ratio | 2.698315 | p-value | 0.03141 |
| Neutrophils | | not selected | selected |
| TFBS | not selected | 1749 | 12 |
| | selected | 234 | 5 |
| odds ratio | 3.111739 | p-value | 0.04318 |
| Monocytes | | not selected | selected |
| TFBS | not selected | 1739 | 17 |
| | selected | 237 | 7 |
| odds ratio | 3.018929 | p-value | 0.02044 |
| Clinically correlated set | | | |
| CD4 T cells | | not selected | selected |
| TFBS | not selected | 1729 | 27 |
| | selected | 235 | 9 |
| odds ratio | 2.450937 | p-value | 0.03366 |
| Neutrophils | | not selected | selected |
| TFBS | not selected | 1744 | 19 |
| | selected | 230 | 7 |
| odds ratio | 2.791547 | p-value | 0.0272 |

# Author Contributions

**Conceptualization:** Y-h. Taguchi, Shohei Komaki, Yoichi Sutoh, Hideki Ohmomo, Yayoi Otsuka-Yamasaki, Atsushi Shimizu.

**Formal analysis:** Y-h. Taguchi.

**Investigation:** Y-h. Taguchi, Shohei Komaki, Yoichi Sutoh, Hideki Ohmomo, Yayoi Otsuka-Yamasaki, Atsushi Shimizu.

**Writing – original draft:** Y-h. Taguchi.

**Writing – review & editing:** Y-h. Taguchi, Shohei Komaki, Yoichi Sutoh, Hideki Ohmomo, Yayoi Otsuka-Yamasaki, Atsushi Shimizu.

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
