## [Decision Letter · Decision Letter 0]

24 Apr 2023

PONE-D-22-35001Integrated analysis of  human DNA methylation, gene expression, and genomic variation in iMETHYL using kernel tensor decomposition-based unsupervised feature extractionPLOS ONE

Dear Dr. Taguchi,

Thank you for submitting your manuscript to PLOS ONE. After careful consideration, we feel that it has merit but does not fully meet PLOS ONE’s publication criteria as it currently stands. Therefore, we invite you to submit a revised version of the manuscript that addresses the points raised during the review process. Please submit your revised manuscript by Jun 08 2023 11:59PM. If you will need more time than this to complete your revisions, please reply to this message or contact the journal office at plosone@plos.org. Please include the following items when submitting your revised manuscript:A rebuttal letter that responds to each point raised by the academic editor and reviewer(s). You should upload this letter as a separate file labeled 'Response to Reviewers'.A marked-up copy of your manuscript that highlights changes made to the original version. You should upload this as a separate file labeled 'Revised Manuscript with Track Changes'.An unmarked version of your revised paper without tracked changes. You should upload this as a separate file labeled 'Manuscript'.

We look forward to receiving your revised manuscript.

Kind regards,

Turki Talal Turki, Ph.D.

Academic Editor

PLOS ONE

Journal Requirements:

3. Thank you for stating the following financial disclosure: "No"

Reviewers' comments:

Reviewer's Responses to Questions

**Comments to the Author**

1. Is the manuscript technically sound, and do the data support the conclusions?

Reviewer #1: Yes

Reviewer #2: No

Reviewer #3: Yes

2. Has the statistical analysis been performed appropriately and rigorously? 

Reviewer #1: Yes

Reviewer #2: Yes

Reviewer #3: Yes

3. Have the authors made all data underlying the findings in their manuscript fully available?

Reviewer #1: Yes

Reviewer #2: Yes

Reviewer #3: Yes

4. Is the manuscript presented in an intelligible fashion and written in standard English?

Reviewer #1: Yes

Reviewer #2: Yes

Reviewer #3: Yes

5. Review Comments to the Author

Reviewer #1: Comments to the Author

The given manuscript entitled, “Integrated analysis of human DNA methylation, gene expression, and genomic variation in iMETHYL database using kernel tensor decomposition-based unsupervised feature extraction" integrated the gene expression, DNA methylation, and genome variants for 194 subjects; divided as CD4+ T cells (n=99), monocytes (n=99) and neutrophils (n=94) from the iMETHYL database using the kernel tensor decomposition-based unsupervised feature extraction with very limited computational resources. The work comprised the analysis of data sets from these 194 subjects from iMETHYL database. Further, Pre-processing of data-sets comprised of: Fastq files from RNA-Seq were processed through the adapted GTEx pipeline V8; alignment of sequence reads to the GRCH37 human genome using STAR v2.5.0. Also, the sequence reads from whole-genome bisulphite sequencing were aligned using NovoAlignV3.02.08 etc. tensor decomposition-based unsupervised feature extraction were applied and genes associated with selected genomic regions were identified using R. It was followed by enrichment analysis and TF regulation analysis and TFBSs and genes associated with variants were identified.

Based on this background, study results, this reviewer feels that the manuscript writing and methodology adopted needs more clarity and needs certain improvement in the light of following comments and can be accepted after minor revisions for the esteemed journal PLOS ONE.

Minor Criticism:

A. Abstract & title & Aim :

The title should be improved. Word iMETHYL does not convey its true essense . Since iMETHYL is a database, iMETHYL should be replaced with iMETHYL database. The authors should be very careful about the title so that it conveys the true essense of the work presented. The statement, “ Our methods do not even require pre-knowledge about the subjects, as they are fully unsupervised.” Seems ambiguous and perhaps need amendment. In the abstract, statement “Integrating gene expression, DNA methylation, and genomic variants …….is impossible” is contradicting the present research. It is also contradicting the further statement in introduction. Kindly update/clarify.

B. Introduction:

Kindly make the sentence in agreement with abstract/ here.

C. Materials and Methods:

1. The web-link for iMETHYL database should be provided.

2. Approval no. from ethics committee should be mentioned.

3. Flow diagram (s) of methodology and pipelines used should be provided.

D. Results: well written

E. Discussion: Adequate.

The use of abbreviation for any terminology should be first expanded on its first use.

The authors are encouraged to improve the manuscript for any typographical and grammatical errors by reading the manuscript word by word.

Reviewer #2: The authors applied tensor decomposition-based unsupervised feature extraction approach to integrate gene expression, DNA methylation, and genome variants and detect joint features. Multiple applications of the methods were used to demonstrate approach. I have the following comments for the authors to consider:

Major comments are in the section of Material and methods.

1) Material and methods:

a. It will be helpful to explain the method with more detailed information to justify the use of u_l2j to detect features and help readers understand the rationale. It seems u_l2j represents loading? The two references the authors cited are both books without specific indications of the sources of the methods, making the understanding of the approach more challenging.

b. The notations of the variables and data are confusing in terms of their dimensionality.

c. The p-value defined in (4) is for testing H0: u_.ik=0. This definition was never used in subsequent analyses and thus unclear about its purpose.

d. In the Introduction section, the authors noted that an existing method did not discuss the relationship between copy number variants and DNA methylation. However, from my understanding, the method applied in the current study was not able to assess the relationship among the three omics either.

e. For genetic variants data, since the values are ordinal with three levels, it is unclear if the proposed decomposition still works and does not violate the assumptions.

2) Results: A lot information is provided in this section. It might be helpful to provide a flow chart of the results to guide readers through.

Minor comment: Lines 24 to 26 in the subsection “Data Set”, it is unclear how the number 194 was reached. Maybe a Venn diagram will help.

Reviewer #3: Integrated analysis of human DNA methylation, gene expression, and genomic

variation in iMETHYL using kernel tensor decomposition-based unsupervised feature extraction by Taguchi et al.

In this article, the authors propose a novel method to be able to integrate multiomic data (DNA methylation, gene expression, and genomic variants) with limited bioinformatics resources despite be non-coincidents (i.e.,the distance between them). They propose a method in which they enrich the data that are connected through regulation by transcription factors. They analyzed data profiles from 194 healthy subjects.

To accomplish with this goal, they used a Tensor decomposition-based unsupervised feature extraction. Authors also explain the step-by-step process to enrich the data and form a biologically significant set.

The manuscript is well written, authors detail each step of the process (i.e. formulas, tables), they have already adjusted the p-values which is a requisite because of the multiple comparisons performed.

I found this paper interesting. I have minor comments:

-This was done with peripheral blood cell data, my question would be if this can be extrapolated, used for other types of cells.

-I could not find demographic characteristics of these samples. I understand that this is not the aim of the manuscript, but I was wondering if the authors adjusted the multiomic data for variables like sex and age?. Please comment on this.

-The sample analyzed by the authors was small (n=194), is this method able to analyze datasets of multiomic data from large cohorts of participants?

-Regarding statistical significance and p-values, please, comment on the effect size (I mean different effect for methylation, gene expression and genetic variants) which is a critical requirement for multiple testing correction on thousands of genes in these multiomic analyzes.

-In table 16, please include the corresponding IC for the OR values.

-it is recommended that the authors comment if there are limitations in their study.

Minor misspelling: -in Table 2 “Clincally”

6. PLOS authors have the option to publish the peer review history of their article (what does this mean?). If published, this will include your full peer review and any attached files.

Reviewer #1: **Yes: **Varij Nayan

Reviewer #2: No

Reviewer #3: No

---

## [Decision Letter · Decision Letter 1]

10 Jul 2023

Integrated analysis of  human DNA methylation, gene expression, and genomic variation in iMETHYL database using kernel tensor decomposition-based unsupervised feature extraction

PONE-D-22-35001R1

Dear Dr. Taguchi,

We’re pleased to inform you that your manuscript has been judged scientifically suitable for publication and will be formally accepted for publication once it meets all outstanding technical requirements.

Kind regards,

Turki Talal Turki, Ph.D.

Academic Editor

PLOS ONE

Additional Editor Comments (optional):

Reviewers' comments:

Reviewer's Responses to Questions

**Comments to the Author**

1. If the authors have adequately addressed your comments raised in a previous round of review and you feel that this manuscript is now acceptable for publication, you may indicate that here to bypass the “Comments to the Author” section, enter your conflict of interest statement in the “Confidential to Editor” section, and submit your "Accept" recommendation.

Reviewer #1: All comments have been addressed

Reviewer #2: All comments have been addressed

Reviewer #3: All comments have been addressed

2. Is the manuscript technically sound, and do the data support the conclusions?

Reviewer #1: Yes

Reviewer #2: Yes

Reviewer #3: Yes

3. Has the statistical analysis been performed appropriately and rigorously? 

Reviewer #1: Yes

Reviewer #2: Yes

Reviewer #3: Yes

4. Have the authors made all data underlying the findings in their manuscript fully available?

Reviewer #1: Yes

Reviewer #2: Yes

Reviewer #3: Yes

5. Is the manuscript presented in an intelligible fashion and written in standard English?

Reviewer #1: Yes

Reviewer #2: Yes

Reviewer #3: Yes

6. Review Comments to the Author

Reviewer #1: Since the authors have addressed the concerns and updated the manuscript adequately, manuscript may be accepted.

Reviewer #2: Thanks to the authors to address the comments/suggestions raised in the previous round. Figure 2 is quite informative.

Reviewer #3: After reading the authors' responses to the three reviewers and the modifications (including flowcharts, diagrams) that the authors have made to the manuscript based on our observations, I consider that the article can be accepted for publication in its current form.

Please, genes names should be italicized in Tables 4 & 6.

7. PLOS authors have the option to publish the peer review history of their article (what does this mean?). If published, this will include your full peer review and any attached files.

Reviewer #1: **Yes: **Varij Nayan

Reviewer #2: No

Reviewer #3: No

---

## [Editor Report · Acceptance letter]

14 Jul 2023

PONE-D-22-35001R1 

Integrated analysis of  human DNA methylation, gene expression, and genomic variation in iMETHYL database using kernel tensor decomposition-based unsupervised feature extraction 

Dear Dr. Taguchi:

I'm pleased to inform you that your manuscript has been deemed suitable for publication in PLOS ONE. Congratulations! Your manuscript is now with our production department. 

Kind regards, 

on behalf of

Dr. Turki Talal Turki 

Academic Editor

PLOS ONE